# Monitoring of Measurable Residual Disease Using Circulating DNA after Allogeneic Hematopoietic Cell Transplantation

**DOI:** 10.3390/cancers14143307

**Published:** 2022-07-07

**Authors:** Miguel Waterhouse, Sandra Pennisi, Dietmar Pfeifer, Florian Scherer, Robert Zeiser, Justus Duyster, Hartmut Bertz, Jürgen Finke, Jesús Duque-Afonso

**Affiliations:** 1Department of Hematology Oncology and Stem Cell Transplantation, University of Freiburg Medical Center, 79106 Freiburg, Germany; sandra.pennisi@uniklinik-freiburg.de (S.P.); dietmar.pfeifer@uniklinik-freiburg.de (D.P.); florian.scherer@uniklinik-freiburg.de (F.S.); robert.zeiser@uniklinik-freiburg.de (R.Z.); justus.duyster@uniklinik-freiburg.de (J.D.); hartmut.bertz@uniklinik-freiburg.de (H.B.); juergen.finke@uniklinik-freiburg.de (J.F.); 2Faculty of Biology, Albert Ludwigs University of Freiburg, 79104 Freiburg, Germany

**Keywords:** measurable residual disease, mixed chimerism, circulating cell-free DNA, hematological relapse, allogeneic stem cell transplantation, extramedullary relapse

## Abstract

**Simple Summary:**

The major cause of treatment failure after allogeneic stem cell transplantation (allo-HSCT) is due to relapse of the underlying disease. Novel methods and strategies are needed to detect early relapse after allo-HSCT. The present study reports the clinical utility of monitoring measurable residual disease (MRD) and mixed chimerism (MC) by droplet-digital PCR in circulating cell-free DNA (cfDNA) in 62 patients with myeloid malignancies undergoing allo-HSCT. MC in circulating cfDNA at an optimal threshold of 18% discriminated patients with hematological relapse from patients in complete remission after allo-HSCT. Most of the mutations identified using a targeted next-generation sequencing (NGS) panel were detected in cfDNA at relapse and were suitable for the monitoring of MRD. In several cases, mutations were detected earlier in cfDNA than in peripheral blood mononuclear cells. In conclusion, longitudinal analysis of cfDNA for MRD and MC can be used as a complementary tool for early detection of relapse in patients after allo-HSCT and could be used to guide clinical interventions.

**Abstract:**

Relapse of the underlying disease is a frequent complication after allogeneic hematopoietic stem cell transplantation (allo-HSCT). In this study, we describe the clinical utility of measurable residual disease (MRD) and mixed chimerism (MC) assessment in circulating cell-free DNA (cfDNA) analysis to detect earlier relapse in patients with hematological malignancies after allo-HSCT. A total of 326 plasma and peripheral blood mononuclear cell (PBMCs) samples obtained from 62 patients with myeloid malignancies were analyzed by droplet-digital PCR (median follow-up: 827 days). Comparison of MC in patients at relapse and in complete remission identified an optimal discriminating threshold of 18% of recipient-derived cfDNA. After performing a targeted next-generation sequencing (NGS) panel, 136 mutations in 58 patients were detected. In a total of 119 paired samples, the putative mutations were detected in both cfDNA and PBMCs in 73 samples (61.3%). In 45 samples (37.8%) they were detected only in cfDNA, and in only one patient (0.9%) were they detected solely in DNA from PBMCs. Hence, in 6 out of 23 patients (26%) with relapse after allo-HSCT, MRD positivity was detected earlier in cfDNA (mean 397 days) than in DNA derived from PBMCs (mean 451 days). In summary, monitoring of MRD and MC in cfDNA might be useful for earlier relapse detection in patients with myeloid malignancies after allo-HSCT.

## 1. Introduction

The major cause of treatment failure after allogeneic stem cell transplantation (allo-HSCT) is relapse of the underlying disease [1]. Detection of disease recurrence after allo-HSCT allows prompt intervention, which in turn results in improved outcomes [2,3].

Long-standing evidence suggests that early detection of measurable residual disease (MRD) by identifying leukemia-associated immunophenotypes (LAIP) by flow cytometry, and fusion transcripts as well as gene expression (e.g., WT1) by quantitative RT-PCR, are able to predict relapse in patients with acute myeloid leukemia (AML) [4,5,6]. Recent technological advances to detect MRD after allo-HSCT such as multicolor flow cytometry, the introduction of droplet digital PCR (ddPCR) and next-generation sequencing (NGS), have resulted in improved MRD detection [7,8,9,10]. As recently reported by an expert panel, MRD detection in AML is based either on PCR amplification of leukemia-associated targets or on flow cytometric detection of LAIPs. In addition, the same expert panel suggests the use of hematopoietic chimerism as a surrogate method for MRD detection after allo-HSCT [11].

Currently, quantification of AML MRD levels after allo-HSCT and the clinical implications thereof are defined on peripheral blood, bone marrow or isolated cell subsets. Peri-transplant detection of MRD in peripheral blood mononuclear cells (PBMCs) and bone marrow mononuclear cells (BMMCs) after allo-HSCT by flow cytometry, gene expression profiling, quantitative RT-PCR and NGS methods, have been associated with worse outcomes [12,13,14,15,16]. However, a frequent event such as isolated extramedullary relapse cannot, by definition, be detected in PBMCs or BMMCs. Furthermore, there is a lack of an appropriate and accurate method for early detection of extramedullary relapse.

In recent years, we and others have been investigating the potential use of cell-free DNA (cfDNA) isolated from plasma for complication assessment after allo-HSCT [17,18,19,20,21,22]. We were able to show that longitudinal analysis of chimerism in cfDNA can be reliably used for detection of graft-versus-host disease (GvHD) during the follow-up visit after allo-HSCT [22]. By using cfDNA and methylated organ-specific genes, patients responding to GvHD treatment could be identified [22]. These initial studies prompted us to further investigate the use of cfDNA in another relevant allo-HSCT complications, namely relapse of the underlying disease through the detection of mixed chimerism and leukemia-associated genetic aberrations in cfDNA after allo-HSCT. Furthermore, we analyzed MRD and chimerism kinetics in cfDNA in patients relapsing after allo-HSCT.

## 2. Materials and Methods

### 2.1. Patient Samples

A total of 326 peripheral blood (PB) samples obtained from 62 patients diagnosed with myeloid malignancies and undergoing allo-HSCT were included in the study (Table 1). Samples were collected during treatment in the hospital and in the standard routine controls in the outpatient clinic. There were no fixed time points to collect samples from the patients. Nevertheless, a mean of 5.2 samples per patient during the study were collected and analyzed. The median follow-up of the patients was 827 days (range: 52–4363 days) after allo-HSCT (Appendix A). PB samples from donors were not included in the study. Samples from healthy controls were used as negative controls to establish ddPCR assays for chimerism and MRD. The transplantation procedure and GvHD prophylaxis were performed as previously described [23]. Post-transplant events such as hematological relapse, GvHD and infections, among others, were defined based on standard clinical and laboratory criteria. The study was approved by the Ethics Committee of the Albert Ludwigs University of Freiburg, Freiburg, Germany (Nr 471/17). Written informed consent was obtained from the patients in accordance with the declaration of Helsinki.

### 2.2. DNA Isolation

Peripheral blood and bone marrow processing, isolation of PBMCs and BMMCs, plasma isolation and storage were performed as previously described [9]. Genomic DNA was extracted from PBMCs and cfDNA from plasma using the Qiasymphony miniDNA and the QIAmp circulating nucleic acid kit according to the manufacturer’s instructions (Qiagen GmbH, Hilden, Germany).

### 2.3. Next-Generation Sequencing

BMMC or PBMC specimens were analyzed for gene/mutation hotspots using a 54-gene/hotspot targeted NGS panel (whole coding sequences for 15 genes and hotspots for 39 additional genes, TruSight myeloid panel, Illumina) and sequencing was performed using a MiSeq platform (Illumina). Samples were analyzed either at initial diagnosis or in those patients with active disease before allo-HSCT.

### 2.4. Chimerism Testing and Mutational Analysis

With the exception of the FLT3-ITD mutation, both chimerism status and MRD detection in DNA derived from PBMCs and plasma were performed simultaneously and analyzed using the QX200 ddPCR system (Bio-Rad Laboratories, Munich, Germany). Determination of the length of FLT3 insertion fragment and quantification of FLT3-ITD mutation load were performed by fragment analysis (Fragment Analyzer, Agilent Technologies GmbH, Waldbronn, Germany). The panel of insertion/deletion polymorphic markers used for chimerism testing and their interpretation has already been described [7,8]. We have previously shown that patients with either acute or chronic GvHD have increased MC in cfDNA samples [19]. Therefore, patients with active GvHD were excluded from the analysis for chimerism assessment in cfDNA (*n* = 10; total number of patients analyzed for chimerism status *n* = 52).

Mutations associated with myeloid neoplasms and identified using the targeted NGS panel were used as MRD markers. A total of 18 assays were designed for MRD detection by ddPCR (NPM1 mutation type A and B, IDH1 R132C, IDH1 R132H, IDH2 R140Q, NRAS G13C, NRAS G12S, DNMT3A R882H, JAK2 V617F, KRAS Q61R, U2AF1 Q157P, U2AF1 Q157R, FLT3-ITD, Calreticulin 52 base pair deletion, SETBP1 G870S, SF3B1 K700E and MPL S505N). According to the European Leukemia Net (ELN) suggestion, the assays used to detect MRD in our study were able to detect leukemic cells to a level of 0.1% [11]. In this study, MRD positivity was defined as the detection of a residual disease marker above the assay-defined limit of detection. To analyze the mutation load relationship between cfDNA and PBMCs, a mutation ratio between both sample types was calculated using the following formula: mutation ratio = mutation load cfDNA/mutation load PB.

### 2.5. Statistical Analysis

Statistical parameters were calculated using Analyse-it software version 5.51 (Analyse-it, Leeds, UK). Correlation coefficients were calculated by Spearman rank correlation analysis. For the comparison of qualitative or quantitative variables without a normal distribution the Mann–Whitney, Wilcoxon signed-rank or Kruskal–Wallis tests were used. Quantitative variables were analyzed with the Student’s paired t-test or Fisher’s exact test in the case of small numbers [12]. All tests were 2-sided, accepting *p* ≤ 0.05 as indicating a statistically significant difference. The general performance of the assays used for chimerism detection was analyzed by plotting the true-positive rate (sensitivity) and the false-positive rate (1-specificity) in a receiver operating characteristic (ROC) space. The Youden index was used to estimate the optimal threshold value for each assay. The threshold for each molecular marker was established as suggested by the Clinical and Laboratory Standards Institute (CLSI guideline EP 17 A2).

## 3. Results

### 3.1. Chimerism Analysis of cfDNA at Relapse

A total of 281 samples from 52 patients were analyzed for chimerism assessment after excluding patients with active GvHD, which has been found to increase MC in plasma [22]. Relapse of the underlying disease after allo-HSCT was detected in 24 patients. Among this group, extramedullary relapse was detected in three patients (extramedullary relapse site: central nervous system, skin and breast). All patients without hematological relapse, as well as patients with extramedullary relapse showed complete donor chimerism in PBMCs after allo-HSCT at the time points analyzed. Meanwhile, increasing MC was detected in PBMCs in all patients before hematological relapse.

We next analyzed the chimerism status in cfDNA and compared it with PBMCs. The mean percentage of recipient cfDNA from patients in complete remission and without GvHD or other transplant-related complication (*n* = 28) was 6.7% (range: 0–25%), whereas recipient cfDNA in patients with hematological relapse was 47.3% (range: 6–94%). In patients with extramedullary relapse the percentage of recipient cfDNA was 11.3% (range: 6–16%). A significant difference in the recipient-derived cfDNA percentage at relapse when compared with cfDNA from patients in complete remission was observed (*p* < 0.001). In contrast, in those patients with extramedullary relapse no significant difference in recipient-derived cfDNA was observed when compared with patients in complete remission (Figure 1A). The mean percentage of recipient-derived DNA in PBMCs in those patients with hematological relapse was 16.7% (range: 1–82%). The difference between the recipient-derived cfDNA and genomic DNA from PBMCs at this time point was statistically significant (*p*-value < 0.001) (Figure 1B).

After performing the ROC curve, the obtained area under the curve was 0.968 (95% CI; 0.929–1.003, *p*-value < 0.001) and the optimal discriminating threshold between patients at the relapse time point and those in complete remission, was 18% of recipient-derived cfDNA (sensitivity: 93.4% specificity: 90.9%) (Figure 1C,D). The percentage of recipient DNA in paired plasma and PBMCs samples showed no significant correlation (Spearman r = 0.148) (Appendix A).

### 3.2. MRD Monitoring in cfDNA and PBMCs in Patients Relapsing after Allo-HSCT

In our previous work, we showed that recipient DNA or MC could be detected in cfDNA in patients with hematological relapse, aGvHD and other allo-HSCT complications [22]. In order to distinguish patients with relapse from those with other allo-HSCT complications, we established MRD monitoring of leukemia-specific mutations. After performing targeted NGS in patient-derived BMMC or PBMC samples at diagnosis, we were able to detect a total of 136 mutations in 58 patients (Figure 2A,B). The mean mutation number per patient was 2.3 (range: 1–5). Therefore, in 45 out of 58 patients at least one mutation could be used as a MRD marker. For some mutations detected in our targeted NGS panel, a ddPCR assay could not be established with enough sensitivity and specificity for MRD assessment in cfDNA and PBMCs. From the group of 45 patients with an MRD marker, 17 of them developed hematological relapse, whereas 3 patients developed extramedullary relapse after allo-HSCT. The most frequent mutations found by targeted NGS at diagnosis or before allo-HSCT were NPM1 (*n* = 19, 31%), DNMT3A (*n* = 14, 23%) and TET2 (*n* = 10, 16%).

Based on our findings that the plasma from patients relapsing after allo-HSCT showed a significantly higher percentage of recipient-derived cfDNA compared with those patients in complete remission, we hypothesized that part of this recipient-derived cfDNA contained the mutation originally detected in leukemic cells before allo-HSCT when examined by NGS. For this purpose, paired PBMCs and plasma samples were compared at different time points after allo-HSCT. A total of 119 samples from 24 patients were above the limit of detection of the designated mutation. The putative mutation was detected in both cfDNA and PBMCs in 73 samples (61.3%), whereas in 45 samples from 17 patients the mutation was detected only in plasma (37.8%). This would suggest that cfDNA shows higher sensitivity for mutation detection. In the remaining sample (0.9%) the mutation was detected only in PBMCs (Figure 3A).

We analyzed the number of samples per patient with MC or detection of MRD positivity in cfDNA (Appendix A). A significantly higher number of samples in patients with MC in cfDNA were analyzed compared with patients without MC (median number of samples per patient 5.5 vs. 2.5, *p*-value 0.015, Mann–Whitney test). Similarly, more samples were analyzed in patients with MRD positivity in cfDNA compared with patients with MRD negativity (median number of samples per patient 6 vs. 4, *p*-value 0.041, Mann–Whitney test). We hypothesize that patients with complications after allo-HSCT, i.e., molecular/hematological relapse, more often underwent diagnostic measurements in the outpatient clinic.

Interestingly, in 12 samples obtained from the three patients with extramedullary relapse, the MRD mutation could be detected in the cfDNA but not in the PBMCs. In this last group of patients, the internal tandem duplication of the FLT3 mutation (FLT3-ITD) in the DNA obtained from cerebrospinal fluid and the one found in cfDNA showed the same size. When analyzing the mutation tumor load between cfDNA and paired individual PBMC samples, a mean of 2.6-fold difference in cfDNA (range: 1.3–6.1) when compared with PBMCs was found. The mean mutational tumor load grouped by mutated genes in PBMCs was 4.9 (range: 0.2–10) and in cfDNA was 10.7 (range: 0.5–20) (Table 2, Appendix A). Individual mutational tumor load comparison between cfDNA and PBMCs shows significant differences in selected mutations (FLT3-ITD, KRAS, NPM1, DNMT3A), whereas in the rest of the analyzed mutations no significant difference in the tumor load was found (SF3B1, U2AF1, IDH1/2, JAK2, NRAS). With the exception of FLT3-ITD, a moderate to significant correlation in the mutational tumor load between cfDNA and PBMCs in the analyzed mutations was observed (Table 2).

### 3.3. Chimerism and Mutation Kinetics in cfDNA after Transplantation

In twenty-six patients after allo-HSCT, recipient-derived cfDNA above the established chimerism discriminating threshold (18%) was detected before MC could be found in PBMCs. In this last group of patients, the mean detection time of MC for the first time after allo-HSCT was 328 days in cfDNA (range: 26–645 days) and 404 days in PBMCs (range: 48–1098 days). However, this time difference did not reach statistical significance (*p* = 0.31). In the remaining patients (*n* = 15), MC was detected for the first time at the same time point in cfDNA and PBMCs. The mean time for MC detection in this last group was 230 days (range: 27–652 days) (Figure 3B). MC levels in cfDNA from patients in complete remission and without evidence of complications after allo-HSCT remained stable, with moderate intra-individual fluctuations under the established threshold.

In agreement with the findings mentioned above, MRD positivity after allo-HSCT was detected earlier in cfDNA than in PBMCs in six patients. In this case, the mean MRD detection time after allo-HSCT was 247 days (range: 52 to 395 days) in cfDNA, and 534 days in PBMCs (range: 211 to 838 days). This time difference shows a trend for significance (*p* = 0.06). The clinical characteristics of these six patients are shown in Table 3. In the remaining patients (*n* = 17) MRD positivity in PBMCs and cfDNA was detected at the same time point with a mean of 249 days (range: 10 to 652 days) (Figure 3C). We next investigated the correlation between the tumor load measured as MRD and MC in cfDNA in patients with hematological relapse. With the exception of FLT3-ITD, all other tested mutations (IDH2 R140Q, SF3B1 K700E, NRAS G13C, KRAS Q61R) reveal a moderate to significant correlation with MC in cfDNA. NPM1 mutation shows a negative correlation with MC in cfDNA (Appendix A).

Chimerism and MRD kinetics in PBMCs and cfDNA showed different time-course patterns, independent of the chimerism or MRD marker used (Appendix A). Interestingly, in those patients with extramedullary relapse, MRD positivity increased whereas the levels of MC in cfDNA were found to be stable or show only moderate fluctuations (Figure 4A,B).

## 4. Discussion

Early detection of MRD positivity after allo-HSCT prompts clinical decision making and eventually therapeutic intervention to control impending disease relapse. Several studies have shown the clinical utility of MRD detection in PBMCs or BMMCs [24,25,26]. However, few studies have investigated the usefulness of cfDNA for MRD detection after allo-HSCT [20,21,27]. In addition, data comparing PBMCs and/or BMMCs with cfDNA for MRD assessment is scarce [28]. In this study we investigated the potential use of cfDNA for MRD detection and further compared cfDNA utility with PBMCs.

Our results reveal that the MC status in cfDNA in patients with evidence of hematological relapse is significantly higher when compared with patients in complete remission (mean 6.7% vs. 47.3%, *p* < 0.001). We have previously shown that patients with either active colon or liver aGvHD have an increased amount of recipient-derived cfDNA, similar to that of patients with hematological relapse (mean control 6.8% vs. colon GvHD 31.6% vs. liver GvHD 45.9%) [22]. Taking this into consideration, we excluded those patients with active GvHD. However, in the current study we cannot rule out other conditions that result in increased cfDNA MC, although the most likely explanation for the difference in MC between complete remission and relapse samples is the presence of relapsed disease. For example, drug-induced toxicity and secondary neoplasias have been reported to increase patient-derived cfDNA [29,30,31]. Therefore, increasing MC in cfDNA after allo-HSCT should be interpreted with caution, and other potential sources of recipient-derived cfDNA should be considered. Detection of MRD in cfDNA as described here, along with biomarkers for GvHD in cfDNA as methylated genes [22], might help to elucidate the source of MC and assess the risk for either relapse or GvHD.

A relevant point of our study was the observation that increasing MC and MRD positivity could be detected earlier in cfDNA when compared with PBMCs in a proportion of patients. Although this time difference did not reach statistical significance, earlier MC or MRD positivity detection in cfDNA when compared with PBMCs may identify patients with an increased risk of hematological relapse. Furthermore, earlier relapse detection allows prompt clinical intervention (i.e., tapering of the immunosuppression, donor lymphocyte infusions and novel combination therapies such as decitabine and venetoclax), which in turn might result in an improved allo-HSCT outcome. Nevertheless, due to the small size of our patient cohort and a relatively short follow-up period, our results regarding earlier relapse prediction through the use of cfDNA needs further confirmation by additional studies with larger cohorts and longer follow-up times.

Recently, the clinical utility of mutations identified by targeted NGS in circulating cfDNA was compared with those detected in BMMCs in patients with AML [28]. The authors found 39 unique mutations in 28 genes in 22 patients. Interestingly 5 mutations (13%) were only detected in cfDNA and 15 (38%) only in BMMCs. This study has some important differences to our study. BMMCs were employed as the DNA source for the detection of mutations in their study, whereas we used PBMCs from most of the patients. The targeted NGS panel in BMMCs and in cfDNA was performed at two different time points: at diagnosis and in complete remission. In contrast, we analyzed MRD sequentially at several time points after allo-HSCT (mean = 5.2 samples per patient). Moreover, patients with AML were treated with conventional induction chemotherapy and the patients in our study were treated with allo-HSCT. Therefore, and in line with the mentioned study, our results show that MRD monitoring in BMMCs/PBMCs and in cfDNA in patients with myeloid malignancies are complementary tools.

Of note, in those patients with extramedullary relapse after allo-HSCT we detected the same mutation in cfDNA found originally by NGS at diagnosis or before allo-HSCT. This finding might be clinically relevant, since in those patients with extramedullary relapse there is no circulating biomarker reported to date. In addition, the relapse site in this group of patients was localized in different tissues (breast, skin, central nervous system), suggesting that the relapse anatomical site has no relation with the MRD positivity detected in cfDNA. Further studies in the context of multicenter clinical trials are needed to establish the clinical usefulness of MRD detection in cfDNA in patients with extramedullary relapse after allo-HSCT.

No correlation between the MC levels in cfDNA and MRD load was detected in the internal tandem duplication of the FLT3 mutation. This finding suggests that even with MC percentages below the threshold found in the present study, this particular mutation could be efficiently amplified in cfDNA. This observation also challenges our above-mentioned hypothesis that the plasma from patients with a high percentage of recipient-derived cfDNA relapsing after allo-HSCT should contain the mutation originally found at initial diagnosis. A tentative explanation could be that the kinetics of recipient-derived cfDNA and the mutated DNA found in plasma have different dynamics related to the progression of relapse. Prospective studies with fixed sampling time points are needed to clarify this last issue.

Our study has several limitations that should be acknowledged. A majority of the patient cohort received a reduced intensity conditioning regimen. Future studies should also characterize patients undergoing allo-HSCT after treatment with different conditioning regimens (myeloablative). The patient cohort was relatively small and the median follow-up was also relative short. Our studies focused primarily on patients with myeloid malignancies such as AML and MDS. Further studies in patients with lymphoid malignancies and at high risk for extramedullary relapse (diffuse large B-NHL, acute lymphoblastic lymphoma) should be undertaken to establish MC and MRD positivity in cfDNA for the early detection of hematological relapse across disease entities. Prospective studies with larger more homogenous cohorts of patients (f.e. AML in CR previous allo-HSCT) should assess the relapse risk by MC and MRD positivity at different time points. Several mutations might not be able to identify patients with MRD positivity with high confidence in patients due to, for example, the sensitivity of the assay or the clonal evolution of the leukemic clones. Therefore, the combination of MC and different mutations in cfDNA and PBMCs should increase the detection yield of our approach. Lastly, some mutations detected by NGS analysis in recipient samples have been described as clonal hematopoiesis of indeterminate potential (CHIP) (e.g., DNMT3A) and, in exceptional cases, they may be derived from donor hematopoiesis [32].

## 5. Conclusions

Our data suggest that detection of MRD positivity in cfDNA is comparable to detection in PBMCs, and is able to detect extramedullary relapse, reinforcing its clinical utility in patients after allo-HSCT. In brief, longitudinal analysis of cfDNA for MRD and MC detection can be used as a complementary tool for the detection of MRD and MC in PBMCs and to assess the risk of relapse after allo-HSCT and guide clinical interventions. Further studies are needed, in particular to evaluate the potential use of cfDNA as a response-to-treatment biomarker.

## Figures and Tables

**Figure 1 cancers-14-03307-f001:**
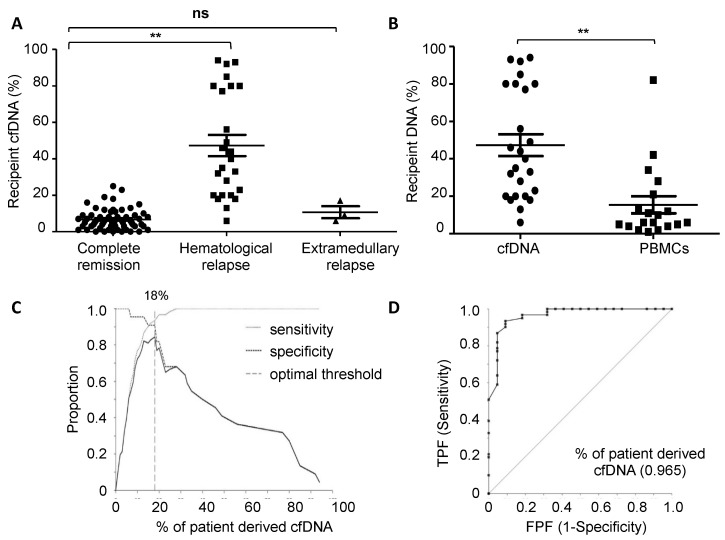
Chimerism assessment in cfDNA after allo-HSCT. (**A**) Graph represents the recipient-derived cfDNA after allo-HSCT in patients with hematological relapse (■), in complete hematological remission (●) and with extramedullary relapse (▲). (**B**) Comparison of recipient-derived DNA in cfDNA (●) and in PBMCs (■) in patients with hematological relapse. Each dot represents a patient, long bars represent the mean and the short bars the standard deviation. ** *p*-value < 0.01, n.s. not significant. (**C**) The resulting area under the curve (AUC: 0.968) revealed cfDNA chimerism analysis as an adequate assay for discriminating relapse from non-relapse. The optimal threshold that discriminates relapse from non-relapse was derived from Youden index and resulted to be 18% of recipient-derived cfDNA. (**D**) To analyze the general performance of cfDNA chimerism assay the true-positive rate (sensitivity) and false-positive rate (1-specificity) was plotted in a receiver operating characteristic (ROC) space. The ROC curve (black line) represents true positive fraction (TPF) against the false positive fraction (FPF). The diagonal (gray line) corresponds to random chance. cfDNA, circulating-free DNA; allo-HSCT, allogeneic hematopoietic stem cell transplantation; PBMCs, peripheral blood mononuclear cells; TPF, true-positive fraction; FPF, false-positive fraction.

**Figure 2 cancers-14-03307-f002:**
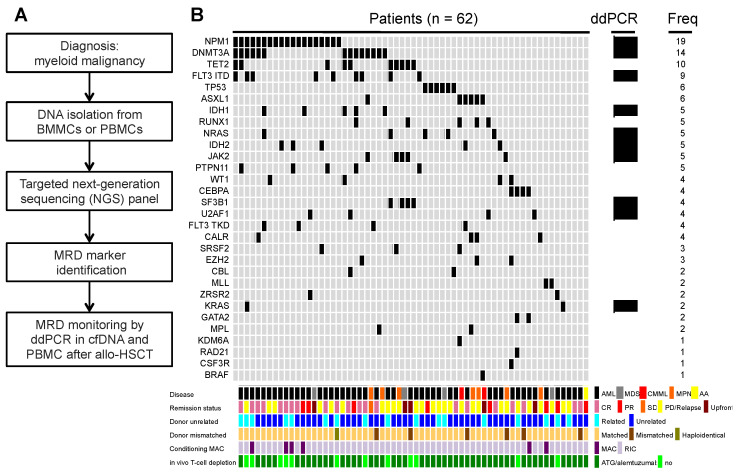
Identification of mutations for MRD monitoring in patients after allo-HSCT. (**A**) Workflow for identification of mutations for MRD monitoring in cfDNA in BMMCs from bone marrow aspirates or in PBMCs from peripheral blood. (**B**) Upper grid panel showing the identified mutations in single patients for MRD monitoring using droplet-digital PCR (ddPCR). Lower grid panel, disease and transplantation characteristics for each individual patient. ddPCR, droplet-digital PCR; cfDNA, circulating-free DNA; MRD, measurable residual disease; allo-HSCT, allogeneic hematopoietic stem cell transplantation; BMMCs, bone marrow mononuclear cells; PBMCs, peripheral blood mononuclear cells; NGS, next-generation sequencing; AML, acute myeloid leukemia; MDS, myelodysplastic syndrome; CMML, chronic myelomonocytic leukemia; MPN, myeloproliferative neoplasia; AA, aplastic anemia; CR, complete remission; PR, partial remission; SD, stable disease; PD, progressive disease; MAC, myeloablative conditioning; RIC, reduced intensity conditioning; ATG, anti-thymocyte globulin; Freq, frequency.

**Figure 3 cancers-14-03307-f003:**
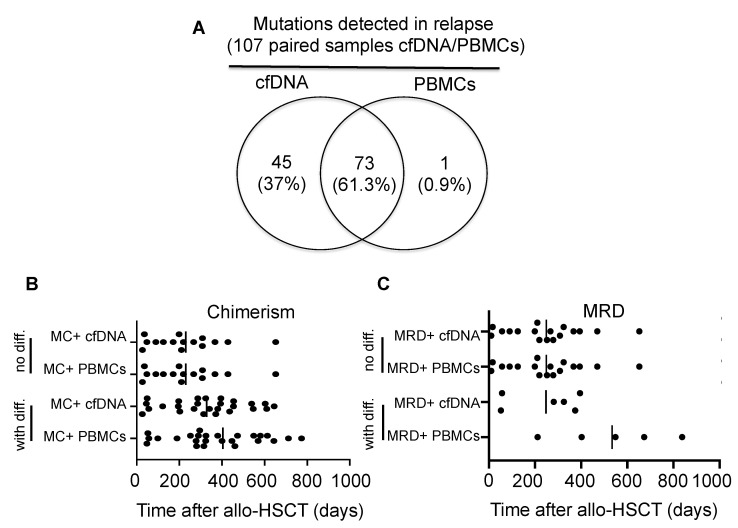
Kinetics of mixed chimerism and measurable residual disease in cfDNA and in PBMCs after allo-HSCT. (**A**) Comparison of mutations detected in paired PBMC and cfDNA samples (*n* = 119) included in this study. Most of the mutations were detected in both, PBMCs and cfDNA (*n* = 73, 61.3%) and a part of the mutations only in cfDNA but not in PBMCs (*n* = 45, 37%). (**B**,**C**) Graphs represent the time to first detection of MC and MRD after allo-HSCT in patients, respectively. Groups of patients were divided, if MC or MRD was detected simultaneously in paired samples (no diff.) or consecutively (with diff.). One patient died because of pneumonia (non-relapse mortality) with MRD detected in cfDNA but not in PBMCs. cfDNA, circulating-free DNA; PBMCs, peripheral blood mononuclear cells; allo-HSCT, allogeneic hematopoietic stem cell transplantation; MRD, measurable residual disease; MC, mixed chimerism; no diff, no difference; with diff., with difference.

**Figure 4 cancers-14-03307-f004:**
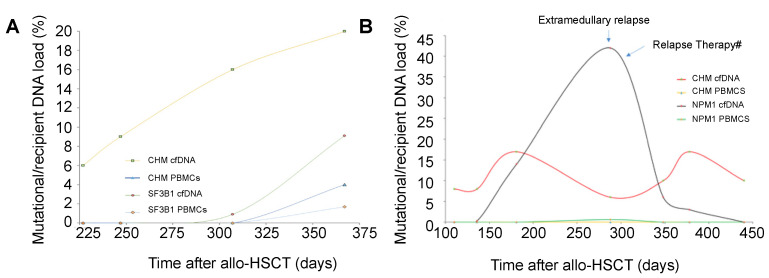
Kinetics of measurable residual disease in cfDNA and PBMCs. (**A**) Representative example of the MRD kinetics of hematological relapse kinetics in an AML patient after allo-HSCT. Increasing MC and MRD (MRD marker: SF3B1 R625C) could be detected earlier in cfDNA (●■) when compared with PBMCs (▲♦). (**B**) In a patient with extramedullary relapse (breast), the MRD marker (NPM1) * increased (▲) and after relapse therapy, a decrease in MRD could be observed. In the same patient a fluctuating MC could be detected during this period (■). cfDNA, circulating-free DNA; PBMCs, peripheral blood mononuclear cells; MRD, measurable residual disease; MC, mixed chimerism; allo-HSCT, allogeneic hematopoietic stem cell transplantation. * NPM1 NCN was multiplied by 100 to fit the scale. #Relapse therapy included local radiation, decitabine and venetoclax.

**Table 1 cancers-14-03307-t001:** Clinical characteristics of patients and transplant.

**Mean follow-up in days (range)**	827 (52–4363)
**Gender male/female**	37/25
**Mean age at transplant (range)**	57 (21–76)
Disease	
AML	48
MDS	4
MPN	7
CMML	2
AA	1
**Karyotype**	
Normal	29
Complex	24
No data	9
**Remission status at allo-HSCT**	
CR	21
Non-CR	41
- Partial remission	8
- Stable disease	1
- Progressive disease/relapse	27
- Upfront	5
**Karnofsky index (%)**	
100	8
90	21
80	14
≤70	19
**Donor female/recipient male**	10
**HLA mismatch**	
Yes	7
No	55
**Donor type**	
Related	17
Non related	45
**Conditioning regimen**	
Myeloablative	5
Reduced toxicity	57
**GvHD prophylaxis**	
CyA/MMF/ATG	44
CyA/MMF	10
CyA/MTX	1
Everolimus/MMF/ATG	5
CyA/MMF/cyclophosphamide	1

AML, acute myeloid leukemia; MDS, myelodysplastic syndrome; MPN, myeloproliferative neoplasia; CMML, chronic myelomonocytic leukemia; AA, aplastic anemia; CR, complete remission; HLA, histocompatibility leucocyte antigen; GvHD, graft-versus-host disease; CyA, cyclosporine A; MMF, mycophenolate mofetil; MTX, methotrexate; ATG, anti-thymocyte globulin.

**Table 2 cancers-14-03307-t002:** Mutation tumor load in PBMCs and cfDNA.

Mutation	N ^1^	Tumor Load PBMCs	Tumor Load cfDNA	*p*-Value
FLT3-ITD	20	0.2	0.5	0.01
KRAS	11	3.3	20	0.0014
NPM1	28	1.5	5.4	0.002
DNMT3A	7	6.8	7.6	0.02
SF3B1	9	3.3	4.2	0.4
IDH1/2	22	5.9	9.7	0.3
JAK2	8	10	17	0.2
NRAS	6	10	16	0.3
U2AF1	8	3.1	15.8	0.06

N ^1^ refers to the number of samples. KRAS, DNMT3A, SF1B3, IDH1/2, JAK2, NRAS and U2AF1 are expressed as percentage of the mutation. NPM1 is expressed as normalized copy number. FLT3-ITD is expressed as ratio: ITD/WT. PBMCs, peripheral blood mononuclear cells; cfDNA, circulating-free DNA.

**Table 3 cancers-14-03307-t003:** Clinical characteristics of patients with MRD positivity in cfDNA before MRD detection in PBMCs after allo-HSCT.

Patient	Age at Allo-HSCT (Years)	Disease	Donor	Conditio-Ning	MRD Pos. in cfDNA (Days)	MRD Pos. in PBMCs (Days)	Hematological Relapse (Days)	Outcome
#1	68	tAML	Unrelated	Reduced	325	673	802	Death after 2. allo-HSCT
#2	23	AML	Unrelated	Reduced	374	402	877 (skin)	Alive after 3. allo-HSCT
#3	59	AML	Related	Myeloa- blative	57	211	1243	Alive after 2. allo-HSCT
#4	60	sAML	Related	Reduced	281	838	366 (CNS-relapse)	Death after after 2. allo-HSCT
#5	52	OMF	Unrelated	Reduced	52	Not detected	n.a.	Non-relapse mortality (pneumonia)
#6	66	CMML	Related	Reduced	395	549	No hematological relapse	Alive at last follow-up

allo-HSCT, allogeneic hematopoietic stem cell transplantation; MRD, measurable residual disease; cfDNA, circulating cell-free DNA; PBMCs, peripheral blood mononuclear cells; tAML, therapy-related acute myeloid leukemia; sAML, secondary acute myeloid leukemia; CNS, central nervous system; OMF, osteomyelofibrosis; CMML, chronic myelomonocytic leukemia.

## Data Availability

The datasets generated during and/or analyzed during the current study are available upon reasonable request from the corresponding authors.

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
