# Peer review of "Monitoring of Measurable Residual Disease Using Circulating DNA after Allogeneic Hematopoietic Cell Transplantation"

_cancers, 2022, doi:10.3390/cancers14143307_

Round 1

Reviewer 1 Report

The manuscript entitled “Monitoring of measurable residual disease using circulating DNA after allogeneic hematopoietic cell transplantation” by Waterhouse et al is interesting. They use MRD and Mixed chimerism in patients with myeloid malignancies undergoing allo-HSCT to address the question of early detection of relapse however there are some critical points that need to be changed.

  • The description of the results is complicated. It is not clear when the samples (time point) were taken and from how many patients at each time point. The description of different patients’ samples in each analysis is a bit confusing. A table with all the details would probably help to address this issue.

  • They have not included to their analysis normal donors as a controls.

  • There have also included different type of the disease which probably makes the results more confusing and difficult to follow.

  • In Figure 2A the authors explained that they monitored MRD in ctDNA however in the workflow there is only bone marrow biopsy. Which one is true?

Minor

  • The explanation of the abbreviations must be clarified the first time they are mentioned in the text. However, in the manuscript some abbreviations such as graft-versus-host disease were explained multiple times in the text (no reason for that).

Reviewer 2 Report

Circulating DNA is a popular concept for the diagnosis and follow-up of solid tumors. Leukemia behaves differently from solid tumors, even in complete remission there may be nests of tumors in sanctuaries or there may be sampling artefacts. As the authors describe themselves graft versus host disease may cause a release of DNA from the recipient. The argument that they chose patients without graft versus is not fully valid because a successful allogeneic transplant always has a certain degree of GVH activity (unless the donor is an identical twin). Nevertheless the concept is interesting- a larger study including myeloablative transplants would be needed.

Suggestions:

1) The authors should explain why they chose a cutoff of 18%

2) The authors should show a dilution series where donor plasma is mixed with recipient plasma

3) The authors should create a Table which shows for the 6 patients in whom circulating DNA preceded clinical relapse:

Age-type of transplant- conditioning-time course of relapse, outcome

Reviewer 3 Report

In this paper, Waterhouse and colleagues investigated the clinical utility of measurable residual disease (MRD) and mixed chimerism (MC) assessment in circulating cell-free DNA (cfDNA) obtained from patients with myeloid neoplasms undergoing allo-HSCT. They analyzed a total of 326 plasma and peripheral blood mononuclear cells (PBMCs) samples obtained from 62 patients by droplet-digital PCR. The MC status in cfDNA in patients with hematological relapse was significantly higher compared with patients in complete remission. In approximately one quarter of patients with relapse after allo-HSCT, MRD positivity could be detected in cfDNA at an earlier time point than in DNA derived from PBMCs. This finding is interesting, although it should be noted that the difference did not reach statistical significance. The authors have recognized this and other limitations of their study in the discussion. Overall, the results of this study are valuable and warrant confirmation in larger patient cohorts. I have some comments.

Major comments

1. A potential issue with this study is that a variable number of samples per patient was studied, and this might skew the results towards cases/mutations for which a larger number of samples was available. In this regard, where there any substantial differences in the number of samples available/studied in cases that show for example an increase in recipient cfDNA, or increased cfDNA mutation load for a specific gene?

2. The data in Table 2 should be presented in a figure as dot plots, showing the distribution of all values in the two types of samples (PBMCs and cfDNA) for each mutated gene. This would be much more informative.

Minor comments

1. What is "Others" under the heading “Disease” in Table 1? As it is only one case, the exact condition should be specified.

2. In Figure 2B, “SF1B3” should be corrected to “SF3B1”.

3. Page 7, Lines 20-21: it should be clarified here why some mutations could not be used as MRD markers for the other 13 patients.

4. Page 7, Lines 24-25: the percentages given for NPM1, DNMT3A and TET2 do not seem to be correct. In Figure 2B, these figures are the number of cases (out of 62), not the percentage.

5. Page 13, Lines 22-23: the use of the word “significant” in the sentence “in a significant proportion of patients,” should be avoided.

6. There are few corrections/amendments that should be made in the text:

- Page 1, Line 37: it is more correct to use the wording “targeted next-generation sequencing (NGS) panel” than “targeted panel next-generation sequencing (NGS)”.

- Page 2, Lines 5-7: The first sentence should read “The major cause of treatment failure after allogeneic stem cell transplantation (allo-HSCT) is relapse of the underlying disease”.

- Page 2, Line 11: correct “by identifying of leukemia-associated immunophenotypes” to “by identifying leukemia-associated immunophenotypes”.

- Page 2, Lines 15-16: correct “has resulted” to “have resulted”.

- Page 2, Line 17: “acute myeloid leukemia” should be spelt in full in line 13, rather than here.

- Page 2, Line 37: in the sentence “These initial studies inspired…”, I would suggest using the word “prompted” instead of “inspired”.

- Page 2, Line 45-46: amend “from 62 patients undergoing allo-HSCT, diagnosed with myeloid malignancies,” to “from 62 patients diagnosed with myeloid malignancies, undergoing allo-HSCT,”.

- Page 11, Lines 26-27: amend the sentence “MC was detected for the first time was at the same time point in cfDNA and PBMCs” to “MC was detected for the first time at the same time point in cfDNA and PBMCs”.

- Page 14, Line 34: correct “maybe” to “may be”.

Author Response

Point-by-point Response Letter

Reviewer #3

Comments and Suggestions for Authors

In this paper, Waterhouse and colleagues investigated the clinical utility of measurable residual disease (MRD) and mixed chimerism (MC) assessment in circulating cell-free DNA (cfDNA) obtained from patients with myeloid neoplasms undergoing allo-HSCT. They analyzed a total of 326 plasma and peripheral blood mononuclear cells (PBMCs) samples obtained from 62 patients by droplet-digital PCR. The MC status in cfDNA in patients with hematological relapse was significantly higher compared with patients in complete remission. In approximately one quarter of patients with relapse after allo-HSCT, MRD positivity could be detected in cfDNA at an earlier time point than in DNA derived from PBMCs. This finding is interesting, although it should be noted that the difference did not reach statistical significance. The authors have recognized this and other limitations of their study in the discussion. Overall, the results of this study are valuable and warrant confirmation in larger patient cohorts. I have some comments.

Major comments

  1. A potential issue with this study is that a variable number of samples per patient was studied, and this might skew the results towards cases/mutations for which a larger number of samples was available. In this regard, where there any substantial differences in the number of samples available/studied in cases that show for example an increase in recipient cfDNA, or increased cfDNA mutation load for a specific gene?
  • Answer: we thank the reviewer for this comment. We analyzed the number of samples per patient with mixed chimerism (cut-off 18%) or detection of MRD positivity in cfDNA. More samples in patients with mixed chimerism (≥18%) in cfDNA were analyzed compare to patients below the defined threshold (median number of samples per patient 5.5 vs. 2.5, p-value 0.015, Mann-Whitney test). Similarly, more samples were analyzed in patients with MRD positivity in cfDNA compared to patients with MRD negativity (median number of samples per patient 6 vs. 4, p-value 0.041, Mann-Whitney test). We interpret that patients with complications after allo-HSCT as, in this case, molecular or hematological relapse were more frequently controlled in the outpatient clinic and underwent more often diagnostic procedures. A comment was added in page 7, lines 11-19 and a supplementary Figure S3 was added.
  1. The data in Table 2 should be presented in a figure as dot plots, showing the distribution of all values in the two types of samples (PBMCs and cfDNA) for each mutated gene. This would be much more informative.
  • Answer: as suggested by the reviewer #3, we show now the results of Table 2 as dot plots in Suppl. Figure 4 for each specific gene in the two type of samples (PBMCs and cfDNA). The Suppl. Figure 4 is cited in the main text (page 8, line 9).

Minor comments

  1. What is "Others" under the heading “Disease” in Table 1? As it is only one case, the exact condition should be specified.

  • Answer: it was a patient with aplastic anemia. The table was corrected as suggested.
  1. In Figure 2B, “SF1B3” should be corrected to “SF3B1”.
  • Answer: SF3B1 was corrected as suggested.
  1. Page 7, Lines 20-21: it should be clarified here why some mutations could not be used as MRD markers for the other 13 patients.
  • Answer: For some mutations detected in our targeted next-generation sequencing (NGS) panel, a ddPCR assay could not be established with enough sensitivity and specificity for MRD assessment in cfDNA and PBMCs. A comment was added (page 6, line 9-11).
  1. Page 7, Lines 24-25: the percentages given for NPM1, DNMT3A and TET2 do not seem to be correct. In Figure 2B, these figures are the number of cases (out of 62), not the percentage.
  • Answer: we thank the Reviewer for this comment. We agree with the reviewer, that the % are actually the number of patients with the mutations. We corrected this and added % of mutations out of the total number of patients (page 6, lines 14-15).

  1. Page 13, Lines 22-23: the use of the word “significant” in the sentence “in a significant proportion of patients,” should be avoided.
  • Answer: we corrected this sentence as suggested (page 11, line 2)

  1. There are few corrections/amendments that should be made in the text:

- Page 1, Line 37: it is more correct to use the wording “targeted next-generation sequencing (NGS) panel” than “targeted panel next-generation sequencing (NGS)”.

  • Answer: We corrected the wording “targeted NGS panel” in several parts of the ms as in page 1, line 8 (Simple Summary); page 1, line 22 (abstract); page 11, line 12; page 11, line 18 and in Figure 2A.

  • Page 2, Lines 5-7: The first sentence should read “The major cause of treatment failure after allogeneic stem cell transplantation (allo-HSCT) is relapse of the underlying disease”.

  • Answer: the sentence was corrected as suggested.

  • Page 2, Line 11: correct “by identifying of leukemia-associated immunophenotypes” to “by identifying leukemia-associated immunophenotypes”.

  • Answer: The sentence was corrected as suggested.

  • Page 2, Lines 15-16: correct “has resulted” to “have resulted”.

  • Answer: The sentence was corrected as suggested.

  • Page 2, Line 17: “acute myeloid leukemia” should be spelt in full in line 13, rather than here.

  • Answer: it was corrected as suggested.

  • Page 2, Line 37: in the sentence “These initial studies inspired…”, I would suggest using the word “prompted” instead of “inspired”.

  • Answer: “inspired” has been replaced by “prompted” as suggested.
  • Page 2, Line 45-46: amend “from 62 patients undergoing allo-HSCT, diagnosed with myeloid malignancies,” to “from 62 patients diagnosed with myeloid malignancies, undergoing allo-HSCT,”.

  • Answer: the sentence was corrected as suggested.

  • Page 11, Lines 26-27: amend the sentence “MC was detected for the first time was at the same time point in cfDNA and PBMCs” to “MC was detected for the first time at the same time point in cfDNA and PBMCs”.

  • Answer: the sentence was corrected as suggested (page 9, line 8).

  • Page 14, Line 34: correct “maybe” to “may be”.

  • Answer: it was corrected as suggested.

Round 2

Reviewer 1 Report

The authors answer the comments, however they did not included cfDNA from healthy donors and the cohort of patients, the time points of the evaluation as well as the number of patients' samples are so heterogenous that any secure conclusion is very difficult to be extracted.

Author Response

The authors answer the comments, however they did not included cfDNA from healthy donors and the cohort of patients, the time points of the evaluation as well as the number of patients' samples are so heterogenous that any secure conclusion is very difficult to be extracted.

-> Answer: we agree with the reviewer that the time points and patient’s samples are very heterogenous and that our study has several limitations, which are addressed in the discussion. This study was performed with real-world data with visits of the patient to the outpatient clinic in the routine checks and not for this study. We think also that with more samples at fixed time points probably MC and MRD positive patients might be detected earlier in cfDNA in PBMCs. 

Reviewer 2 Report

Manuscript is improved, interesting topic, I doubt that circulating DNA will become standard of care any time soon, threshold of 18% still appears somewhat arbitrary

Author Response

Response to Reviewer #2

Manuscript is improved, interesting topic, I doubt that circulating DNA will become standard of care any time soon, threshold of 18% still appears somewhat arbitrary

- Answer: Thank you for the comments for our manuscript. We agree with the Reviewer that 18% cut-off for mixed chimerism identified patients with relapse vs complete remission in our cohort of patients. In our previous publication, a cut-off of 10% mixed chimerism could identify patients with aGvHD colon or aGvHD liver (Waterhouse et al., BMT 2021). These cut-offs should be validated in larger cohort of patients and in prospective study analysis.